



*Manuscript — technical note*

# 2   Analysis of oxygen isotopes of inorganic phosphate
# 3   ($\delta^{18}O_P$) in freshwater: A detailed method description

**Catharina Simone Nisbeth [1], Federica Tamburini [2], Jacob Kidmose [3], Søren Jessen [1] and David**
**William O'Connell [4],***
[1]  Department of Geosciences and Natural Resource Management (IGN), University of Copenhagen, Øster
Voldgade 10, 1350 Copenhagen K, Denmark
[2]  Institute of Agricultural Sciences, ETH Zurich, 8315 Lindau, Switzerland.
[3]  Geological Survey of Denmark and Greenland (GEUS), Øster Voldgade 10, 1350 Copenhagen K, Denmark
[4]  Department of Civil, Structural and Environmental Engineering, Trinity College Dublin, College Green,
Museum Building, Dublin 2, Ireland.
\*  Correspondence: oconnedw@tcd.ie (D.W.O.)
**Abstract:** The ability to identify the origin of phosphorus is essential to effectively mitigate
eutrophication of freshwater ecosystems. The oxygen isotope composition of orthophosphate ($\delta^{18}O_P$)
has been suggested to have a significant prospective as a tracer for P entering freshwater ecosystems.
The $\delta^{18}O_P$ tracing method is, however, still in its preliminary stages and has proven challenging to
implement for new practitioners. In order to achieve progress in developing the application of $\delta^{18}O_P$
signatures as a tracing tool, there is a need to eliminate the methodological challenges involved in
accurately determining $\delta^{18}O_P$. This technical note describes the various steps needed to concentrate
and isolate orthophosphate in freshwater samples into an adequately pure analyte ($Ag_3PO_4$), without
isotopic alteration during processing. The protocol compiles the disperse experiences from previous
studies, combined with our own experience.





## 1. Introduction

In freshwater ecosystems, phosphorus (P) is usually the primary limiting nutrient for growth of macrophytes, algae and cyanobacteria. Increased P concentrations can therefore result in eutrophication, anoxia and degradation of water quality in lakes, rivers and streams (Blake et al., 2005; Hecky & Kilham, 1988; Wetzel, 2001). Phosphorus input to surface water aquatic ecosystems originates from various sources including septic tanks, waste water treatment plants, agricultural fertilizers, animal excreta and dissolved minerals (Heiberg et al., 2012; Marion et al., 1994; Quinton et al., 2010; Sharpley et al., 2003). Thus, identifying the various potential phosphorus sources and their relative contribution to the total phosphorus load is essential for restoration and improvement of eutrophic aquatic ecosystems (Elsbury et al., 2009; McLaughlin, Kendall, et al., 2006).

Identification of the source and apportioning the contributions of phosphorus discharging to surface water from various sources is not a trivial matter and requires an appropriate tracer, which can accommodate this complexity (Jaisi et al., 2011). An ideal tracer is part of the phosphate molecule without changing its properties.

Dissolved inorganic orthophosphate (referred to as $P_i$ hereafter) is the primary form of P cycled through ecosystems (Moorleghem et al., 2013). Hence, the stable oxygen isotope of inorganic phosphate ($\delta^{18}O_P$, in which the subscript 'p' denotes 'phosphate') has been suggested as a significant prospective tracer for P cycling in the environment (Blake et al., 1997, 2005; Colman, 2002; Jaisi & Blake, 2014; McLaughlin et al., 2004). The $\delta^{18}O_P$ can be used as a tracer, since the P-O bond in $P_i$ is resistant to inorganic hydrolysis at temperatures and pH levels found in natural abiotic aquatic ecosystems (Blake et al., 1997; Liang & Blake, 2007; Longinelli et al., 1976). Subsequently, the $\delta^{18}O_P$ value in abiotic aquatic ecosystems will reflect the isotopic signature of the P sources (Tamburini et al., 2010; Zohar et al., 2010). Biological mediation in aquatic ecosystems can, however, alter the source $\delta^{18}O_P$ signatures, through biological uptake and recycling. This will result in an isotopic equilibrium between the stable oxygen isotopes in the ambient water ($\delta^{18}O_w$) and the $P_i$ sources (Blake et al., 2005). Consequently, the $\delta^{18}O_P$ value in abiotic aquatic ecosystems will only reflect the isotopic signature of the P sources when the biological activity is relatively low compared to the input of $P_i$.

### 1.1. The $\delta^{18}O_P$-method

Traditionally, the determination of $\delta^{18}O_P$ was established through fluorination (Crowson et al., 1991; Longinelli, 1966) or bromination of a phosphate precipitate, which generally was in the form of bismuth(III)-phosphate ($BiPO_4$) (Kolodny et al., 1983; Longinelli et al., 1976; Longinelli & Nuti, 1973b, 1973a; Shemesh et al., 1983, 1988). The $BiPO_4$ precipitate is a hygroscopic material that rehydrates within 15 minutes after dehydration, hence significant preparation is required before isotopic analysis. Recent methods use silver(I) phosphate ($Ag_3PO_4$) (Colman, 2002; Crowson et al., 1991; Lécuyer, 2004; Tamburini et al., 2010) which is less hygroscopic, is stabile, has low solubility, and results in better O yield during quantitative conversion of the $PO_4$-O to CO-O, and requires less preparation time (Crowson et al., 1991; Firsching, 1961). Multivalent ions and silicates interfere with $Ag_3PO_4$ precipitation, however low valence ions did not impact precipitation (i.e. $NO_3^-$, $NH_4^+$ and $K^+$) (Firsching, 1961).

Accordingly, $Ag_3PO_4$ precipitation has become the most popular method for $\delta^{18}O_P$ in aqueous and terrestrial environments due to improved extraction protocols enabling sufficient precipitation of $Ag_3PO_4$ for analysis from low inorganic phosphorus concentration matrices (Elsbury et al., 2009; Goldhammer et al., 2011; Granger et al., 2017; McLaughlin et al., 2004; Pistocchi et al., 2017; Tamburini et al., 2010; Zohar et al., 2010). $\delta^{18}O_P$ can by analyzed by thermal conversion/elemental analyser isotope ratio mass spectrometry (TC/EA-IRMS). The $Ag_3PO_4$ precipitation technique for TC/EA-IRMS has many advantages over the traditional fluorination technique in that (*i*) small $PO_4$ quantities are required for the analysis (yielding ~300-600 μg $Ag_3PO_4$); (*ii*) dangerous chemicals are avoided, such as $BrF_5$, $F_2$ or $ClF_3$; and (*iii*) measurements are automated (Vennemann et al., 2002).





*1.2. Approaching a uniform $P_i$ extraction method via $Ag_3PO_4$ precipitation*
Several detailed protocols for the extraction of $P_i$ via precipitation of $Ag_3PO_4$ from different
complex matrix solutions such as fresh and ocean waters and soil extractions exist (Colman, 2002;
Goldhammer et al., 2011; Gruau et al., 2005; McLaughlin et al., 2004; Tamburini et al., 2010, Zohar et
al., 2010). The major techniques for these protocols have been summarized by Paytan & McLaughlin
(2011) and Davies et al. (2014).
For water samples, the broadly common sequence of steps for $Ag_3PO_4$ precipitation is this: (*i*) $P_i$
is quantitatively removed from the sample through magnesium-induced co-precipitation (MagIC) by
brucite (Karl & Tien, 1992); (*ii*) redissolution of the brucite-pallet in an acid matrix, which resuspends
the $P_i$ in solution; (*iii*) removal of other interfering sources of O, such as dissolved organic matter
(DOM), by using anion exchange resins and/or sequential precipitations; (*iv*) removal of potentially
interfering cations using an cation exchange resin; (*v*) precipitation of $Ag_3PO_4$. All steps are designed
to inhibit isotopic fractionation.
One of the major challenges with all the $Ag_3PO_4$ precipitation methods relates to the insufficient
removal of oxygen sources other than phosphate (Tamburini et al., 2010). Thus, the purification steps
are of great importance. Especially DOM is of concern as the high O content of DOM can significantly
interfere with the measured fractionation of $\delta^{18}O_P$ and persists throughout all sequential steps of the
$Ag_3PO_4$ precipitation methods (McLaughlin, Paytan, et al., 2006).
There is a variety of approaches to address this problem, including *a)* adsorption of organic
compounds to phosphate-free activated carbon (Gruau et al., 2005) or to a resin such as DAX-8
(Colman, 2002; Joshi et al., 2018; Tamburini et al., 2010), *b)* repetition of the MagIC step with the
intention of further isolation of $P_i$ from a matrix with potential contaminants (Colman, 2002;
Goldhammer et al., 2011), *c)* acidified pH-specific precipitations of fulvic and/or humic acids (Zohar
et al., 2010), *d)* sequential precipitation and re-crystallization scheme to efficiently scavenge $P_i$
(Tamburini et al., 2010), *e)* and a final washing of the $Ag_3PO_4$ precipitate with hydrogen peroxide to
eliminate residual organic matter by oxidation (Goldhammer et al., 2011; Tamburini et al., 2010;
Zohar et al., 2010).
Despite the several existing protocols and the review papers by Paytan & McLaughlin (2011)
and Davies et al. (2014) focusing on analysis of the $\delta^{18}O_P$ of inorganic phosphate, and despite
numerous articles describing $\delta^{18}O_P$ application in different aquatic environments, there exists
currently no collective or uniform protocol via precipitation of $Ag_3PO_4$ for freshwater matrices. This
is further despite the fact that the method can prove challenging to implement for new practitioners.
In addition, some of the common steps were originally developed and documented for other
conditions than they are now applied on. For example, the MagIC steps' quantitative $P_i$ removal was
well documented, but for the matrix of oceanic seawater, which is relatively invariable compared to
freshwater matrices. Nevertheless, it has nearly directly been applied on freshwater samples. Hence,
to make the method as widely and practically applicable as possible, and to facilitate proper grounds
for a coherent future method development aiming at freshwater systems, there is a need for a detailed
method description for the $Ag_3PO_4$ precipitation method. The present technical note aims to address
the needs with (*i*) describing each step of the $Ag_3PO_4$ precipitation method in detail; (*ii*) explain the
historical background and reasoning behind each step; (*iii*) compile from the literature the (lacking)
documentation of individual steps; (*iv*) give practical advice and suggestions to tackle potential
challenges which may arise when applying the method, as it is, under different scenarios.
**2. Protocol for freshwater $\delta^{18}O_P$ determination**
*2.1. Reading guide for the protocol*
The protocol (Sections 2.2, 2.3 and 2.4) is about concentration and isolation of $P_i$ in freshwater
samples and result in an adequately pure solid silver phosphate crystal ($Ag_3PO_4$), without isotopic
alteration. The description of the subsequent TC/EA-IRMS analysis of the $\delta^{18}O_P$ determination is not
included. For that, we refer to Tamburini et al. (2010) or Davies et al. (2014).





The protocol can be used when water sampling volumes are not restricted. In situations where
sampling is difficult and sample volumes limited, we refer to the method presented by Goldhammer
et al. (2011). If the goal is to determine $\delta^{18}O_P$ from a sediment sample, we refer to the P extraction
method presented by Tamburini et al. (2010).
The here presented protocol consists of three sections: Section 2.2 *'Freshwater sampling'*, Section
2.3 *'Quantitative $P_i$ removal by the MagIC method'*, and Section 2.4 *'Purification and silver phosphate*
*precipitation'*. Each section is divided into main steps presented by roman numerals and each main
step is further subdivided into substeps indicated by letters from the Latin alphabet.
Three different remarks will be presented throughout the protocol:
*Note* ..............................Specific concerns to be aware of when performing one of the substeps.
*We experienced* ..............Phenomena we have experienced that have not been presented in earlier
method descriptions.
*Method disagreement* ....Draws attention to steps where there are inconsistencies between already
published $\delta^{18}O_P$ methods.
The protocol compiles the disperse experiences from previous studies, combined with our own
experience. Description of the preparation of all used chemicals and reagents are provided in
Appendix A.

*2.2. Freshwater sampling*

The amount of water to be sampled depends on the $P_i$ concentration of the sampled water itself.
It is recommended to sample a minimum of 20 μmols of P. This will provide enough P to allow some
P losses from one step to the next and thus an easier handling of the protocol. This can lead to required
water volumes of 10 to 50 L for $P_i$ concentrations of 2 and 0.4 μM, respectively (McLaughlin et al.,
2004; Tamburini et al., 2010).
It is important to take the necessary precaution in relation to the type of water being sampled.
This is especially true when sampling anoxic and $Fe^{2+}$-rich water were $P_i$ co-precipitation with Fe(III)-
(hydr)oxides (henceforth collectively referred to as Fe-oxides), forming upon contact which
atmospheric $O_2$, could immediately occur (Senn et al., 2015). Thus, different sampling approaches are
needed when working with either oxic or anoxic samples:

**Step I. Freshwater sampling**
*a)* Prior to sampling, acid-wash, rinse with deionized distilled water (DD-$H_2O$), and air dry a
polyethylene collection container. If planning to sample anoxic and $Fe^{2+}$-rich water, additionally
flush the container with $N_2$ gas and seal the container. *b)* At the sampling site, fix a piece of nylon
mesh on the opening of the collection container (oxic water sampling) or attach the nylon mesh
to the tip of the sampling tube, submerged in the collection container (ferrous water sampling)
to filter out coarser material. The mesh size depends on practicalities; decide on a size range
which allows a decent flow of water through without clogging. We successfully used a 10 μm
nylon mesh for lake, stream and groundwater, collecting about 1 L per minute using a peristaltic
pump. *c)* Rinse the polyethylene container three times with sampling water before final filling.
When sampling ferrous water, oxygen could enter the anoxic sample and Fe-oxides could start
to precipitate. To avoid those processes rinse and fill the container by pumping water through
the submerged tube into the container and let the water overflow for an extended period of time.
At the final filling, prevent a headspace in the container before closing it. *d)* Collect a parallel
water sample (minimum 10 mL) for measurement of $P_i$ concentration and $\delta^{18}O$ of water, i.e.
$\delta^{18}O_w$.





*Evaluation of the water sampling protocol*

So far, there is no clear guideline regarding the proper filtration requirement for freshwater samples. However, the selected filtration procedure might have an effect on the final obtained purified $Ag_3PO_4$. If necessary particulate organic matter has typically been removed from freshwater samples by filtration through a 0.45 μm GF/F filter (Davies et al., 2014; Elsbury et al., 2009; Li et al., 2011). However, it is extremely impractical to filter many litres of water through a 0.45 micron filter. When working with freshwater samples, filtration of the $HNO_3$ solution after dissolution of the last MagIC pellet (Step VI) could be a solution. Nevertheless, to our knowledge, this still needs to be elucidated further.

*2.3. Quantitative $P_i$ removal by the MagIC method*

The MagIC method was developed by Karl & Tien (1992) and later improved by Thomson-Bulldis & Karl (1998) to precisely determine nanomolar concentrations of SRP and total dissolved phosphorus from marine and freshwater ecosystems. The technique concentrates and isolates $P_i$ from the majority of other dissolved ions, and ideally also from DOP and DOM, thus enabling a more manageable $P_i$ sample for further treatment prior to the final $Ag_3PO_4$ precipitation.

**Step II. Magnesium-induced co-precipitation of dissolved $P_i$ (MagIC)**
Magnesium-induced co-precipitation can quantitatively remove dissolved $P_i$ by adsorption onto $Mg(OH)_2$ (brucite), initiated by addition of NaOH which raises the pH. This is utilized in the first step of the MagIC approach. Brucite can precipitate at any temperature, but temperatures should be kept low (5-10°C) in order to keep microbial activity at a minimum. Microbial activity may alter the source $\delta^{18}O_P$ signatures through biological uptake and recycling (Blake et al., 2005).
The procedure of the brucite precipitation step is as follows:

*a)* Discard some of the sampled water to ensure space for the reactants. *b)* Add 3 M Mg-brine to the water sample in the polyethylene container. The required volume deviate according to the sample volume. Add until the solution achieve a final concentration of ~55 mM $Mg^{2+}$ (Karl & Tien, 1992; Thomson-Bulldis & Karl, 1998); for example, this corresponds to the addition of 1 L 3 M Mg-brine to 50 L freshwater sample. Mix well. The required $Mg^{2+}$ concentration stems from an experimentally evaluated efficacy of $P_i$ removal from $Mg^{2+}$-amended freshwater samples by Karl & Tien (1992) and corresponds to the $Mg^{2+}$ concentration found in seawater. *c)* Then add 1 M NaOH equivalent to 0.5% of the sample solution volume (Thomson-Bulldis & Karl, 1998) and mix again. Check with pH indicator strips that the pH becomes between 9 and 10, as alkaline conditions facilitate brucite precipitation better than acidic conditions (Thomson-Bulldis & Karl,

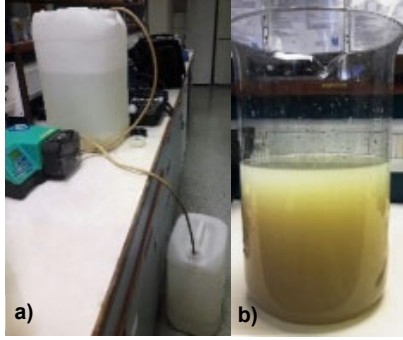

**Figure 1.** (a) Removing the supernatant from the brucite flocs by siphoning, using a peristaltic pump. (b) Brucite flocs left after discarding the supernatant.





1998). If pH <9 add more 1 M NaOH and mix simultaneously. *Note* that excess NaOH does not
improve $P_i$ co-precipitation removal because the resulting higher pH decreases $PO_4$ adsorption.
Rather, excess NaOH has the drawback that it yields a larger mass of brucite flocs which
subsequently must be dissolved in a larger volume of acid (Karl & Tien, 1992). *d)* Allow the
brucite flocs to settle by gravity over a couple of hours. Then remove the supernatant. If the
volume of the sample solution is large, this can be done by siphoning or using e.g. a peristaltic
pump (Figure 1a). The brucite flocs might make up several liters of sludge (Figure 1b). *Note* that
$P_i$ may start to desorb from the brucite flocs, probably because recrystallization of the brucite
lowers the surface area if the suspension is left for longer than it takes the brucite flocs to settle
(Colman, 2002). *e)* Check the absence of $P_i$ in the supernatant, e.g. by using the
spectrophotometric molybdate blue-method (Murphy & Riley, 1986). Discard the supernatant if
$P_i$ has been 100% stripped from the sample solution. If $P_i$ is still present, add additional 1 M
NaOH to the supernatant until no $P_i$ can be detected; the supernatant at this point has a Mg
concentration still matching seawater (supported by PHREEQC modelling). Combine all the
precipitated brucite.
*Method disagreement* regarding the precipitation approach of brucite exists: Joshi et al. (2018) initially
prepared a concentrated MagIC colloidal solution in a split of the sample solution (200–300 mL) and
concurrently adjusted the pH of the remaining sample solution. They then subsequently mixed the
two solutions. The entire volume was then gently shaken continuously to maintain a homogeneous
dispersion of colloids and thus maximize the trapping of $P_i$. Joshi et al. (2018) state that this procedure
is especially prudent when working with low $P_i$ concentrations. This method procedure has
successfully been followed by Yuan et al. (2019). Whether there are discrepancies in the results if one
follows this approach instead of the magnesium-induced approach described in Step II is currently
undocumented.

### Step III. Sample centrifugation

The brucite flocs can be separated from the solution by centrifugation. Do the following:
*a)* After completing Step II, immediately centrifuge the collected brucite floc sludge at 3500 rpm
for 10 minutes, and discard the supernatant. Timewise, it is recommended to use as large
centrifuge tubes as possible, e.g. 250 mL tubes. (No further substeps in Step III.)
*Method disagreement* regarding the recommended centrifugation rotation speed: Karl & Tien (1992)
recommend a low speed (1000 rpm for 1 h) as high g-forces (experienced at >12000 rpm; Karl & Tien,
1992) make the settled brucite flocs harder to dissolve subsequently and do not improve the
separation from the supernatant. In contrast, Goldhammer et al. (2011) recommends a high rotation
speed (10000 rpm for 15 minutes) to ensure complete settling of the fine crystalline $Mg(OH)_2$. The
underlying reasoning for Goldhammer et al.'s approach is that the $\delta^{18}O_P$ of the P trapped in the fine
fraction is significantly different from the $\delta^{18}O_P$ of coarser brucite flocs. In practice, these fines are not
visible to the eye; the supernatant in either case should appear clear. The amount of associated P
therefore must remain tiny compared to the amount in the visibly settled flocs, meaning that the
difference in $\delta^{18}O_P$ needs to be comparably high. Nevertheless, we followed McLaughlin et al. (2004)'s
compromise were a rotational speed of 3500 rpm for 10 minutes was used. This approach was
successful followed by Young et al. (2009) and Elsbury et al. (2009), both working with freshwater
samples. An alternative to centrifugation is gravitational separation used by Colman (2002).

### Step IV. Brucite dissolution

The co-precipitated $P_i$ is re-liberated by dissolving the brucite flocs in 1 M $HNO_3$. The technique is as
follows:
*a)* Add 1 M $HNO_3$ to the centrifuge tubes used for Step III. The required added volume deviate
according to the quantity of brucite flocs. Add until the brucite can be easily removed from the



centrifuge tubes. Be sure to use the minimum amount of acid to minimize acid hydrolysis (Karl
& Tien, 1992); elaborate explanation in section 2.3.1. *b)* Combine the dissolved brucite flocs from
the centrifuge tubes. *c)* Adjust the final pH to ca. 1 using 1 M $HNO_3$ (use indicator pH test strips),
as brucite is first fully dissolved under these conditions; at this point the solution will be liquid
and not viscous.
**Step V. Additional MagIC step**
If the sample contain organic material, the color of the precipitated brucite flocs become tan or even
brown (Goldhammer et al., 2011; Zohar et al., 2010) (Figure 2), whereas it should be milky whitish if
purified (Karl & Tien, 1992). An additional MagIC step, Step V, is thus required leading to (*i*) further
purification of $P_i$ from a matrix with potential contaminants and (*ii*) higher concentrated $P_i$ brucite
flocs (Colman, 2002; Goldhammer et al., 2011). Step V proceeds as follows:
*a)* Raise the pH of the dissolved brucite to about 10-11 by adding 1 M NaOH (do not add the
Mg-brine). Brucite precipitation occurs at pH 9. *b)* Then, repeat Step III and Step IV. *Note* that a
final pH of 1 is still required. *c)* Repeat Step V until discoloration disappears; up to five
repetitions may be necessary (Goldhammer et al., 2011).
*Method disagreement* exists regarding the final pH of the dissolved brucite solution. Colman (2002),
Goldhammer et al. (2011) and McLaughlin et al. (2004) all recommend carefully buffering the solution
back up to a pH between 4 and 6 after re-dissolution of the brucite is complete, making $H_2PO_4^-$ the
main $P_i$ species in the solution. McLaughlin et al. (2004) used 1 M potassium acetate as buffer as it is
inexpensive, nontoxic, and has a low P content, whereas Colman (2002) and Goldhammer et al. (2011)
used 1 M NaOH to adjust pH. However, the subsequent purification steps in these three studies
(precipitation of cerium phosphate (McLaughlin et al., 2004) and a pump-based anion-exchange
chromatography setup (Colman, 2002; Goldhammer et al., 2011)) differ from the purification step
presented in our protocol and thy all utilizes a pH of around 6. In this protocol the subsequent
purification steps utilizes the low pH (see Step VII). Adjustment of pH is therefore not applied in the
MagIC protocol presented in the present study.
**Step VI. Filtration**
After completing Step V one should be left with a solution with a pH of ~1. The final step of the
MagIC protocol separates contaminants insoluble under acid conditions and not incorporated in the
brucite flocs, by vacuum filtration. Do the following:
*a)* Filtrate the dissolved brucite using a 0.7 μm GF/F filter. It may be necessary to centrifuge first
if the floc is not fully dissolved in acid at pH 1. (No further substeps in Step VI.)

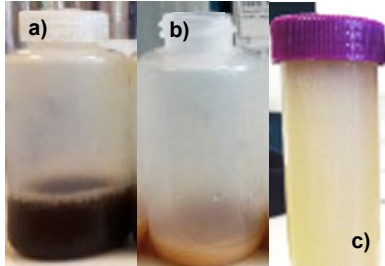

**Figure 2.** Brucite discoloration of sample with high dissolved organic matter (DOM) content. DOM-rich brucite flocs after (a) the first precipitation, (b) after three $HNO_3$ dissolution and NaOH precipitation repetitions and (c) purified brucite (Step V).



After this step, it is recommend to proceed with the the subsequent purification step (Step VII),
without waiting too long. If the samples needs to be stored and/or transported than do not dissolve
the brucite flocs after the last brucite precipitation and store the sample in the fridge. The brucite flocs
should first be dissolve in acid just before the first purification (Step VII).

*Evaluation of the MagIC protocol*

In general, we found the MagIC protocol to be an effective method to remove $P_i$ from water
samples. It is, however, obligatory to check the $P_i$ concentration in the supernatant generated in Step
II before it is discarded.
The MagIC protocol was initially developed for samples of seawater (Karl & Tien, 1992) which
has a nearly constant matrix composition independent of the sampling site and a naturally high
concentration of $Mg^{2+}$ (55 mM). Also, seawater $P_i$ concentrations are rarely high enough to challenge
the quantitative $P_i$ removal in Step II. In contrast to seawater, when working with freshwater samples,
the matrix can vary significantly and $Mg^{2+}$ needs to be added.
Karl & Tien (1992) conducted a limited preliminary investigation of the MagIC technique on
freshwater samples, were the results indicated its applicability. This was later substantiated by
Elsbury et al. (2009), Goldhammer et al. (2011) and McLaughlin et al. (2004). They all used the MagIC
technique for isolation of $P_i$ in freshwater samples. Nonetheless, an initial incomplete quantitative
removal of $P_i$ from the sample solution has been attributed the presence of $HCO_3^-$, as $HCO_3^-$ has an
affinity for brucite similar to $P_i$, and thus reduces the P sorption capacity of brucite (Joshi et al., 2018).
An extra step prior to the MagIC treatment where $HCO_3^-$ is removed by acid treatment forming
degassing $CO_2$ (Joshi et al., 2018) may thus be required; this was not attempted in the present study.
Additional amendments and additions to the MagIC protocol might be necessary when working
with some freshwater samples. This is an important subject, which still needs to be investigated in
more detail.
The acid dissolution of brucite can be a weakness for organic rich samples (i.e. Step IV). During
that step acid hydrolysis may occur, which may potentially convert organic P into new $P_i$ in which
water-O from the ambient environment may be incorporated (McLaughlin et al., 2006). The newly
generated $P_i$ will potentially be incorporated in the $Ag_3PO_4$ crystals, subsequently altering $\delta^{18}O_P$
signature of the sample. By repeating the MagIC step (cf. Step V) the samples are exposed for a longer
acid contact period and thus, there is a higher risk of isotopic alteration driven by acid hydrolysis.
Colman (2002), Thomson-Bulldis & Karl (1998), and Jaisi & Blake (2014) all experimentally
validated, that hydrolysis of a large range of DOP compounds is negligible at extreme (low and high)
pH in the time frame used in routine laboratory processing of samples. Furthermore all reported
hydrolysis impacts on $\delta^{18}O_P$ are below the analytical error (Paytan & McLaughlin, 2011). Yet it is
important to keep in mind the significant variation of the freshwater matrix, and thus the vast array
of organic P compounds with different affinities for the brucite flocs (Colman, 2002; Thomson-Bulldis
& Karl, 1998).
It is therefore wise to test samples that might be susceptible to acid hydrolysis (e.g. organic-rich
samples or samples with an anomalous composition of organic carbon) for isotopic contamination
driven from this mechanism. [18]O-labeled and unlabeled reagents on replicates of the same sample
can be used to trace and correlate the impact of acid hydrolysis.
The positive effect of minimizing the DOM content and other O-bearing compounds remaning
in the sample by repeating the MagIC step might exceed its negative impacts. As mentioned,
inefficient removal of O-bearing contaminating compounds, including DOM, nitrate ($NO_3^-$), sulphate
($SO_4^{2-}$) and calcium carbonate ($CaCO_3$) could result in inclusion of O from other compounds than $PO_4$
in the precipitated $Ag_3PO_4$ (Davies et al., 2014; Lécuyer, 2004) and could thus significantly influence
the measured $\delta^{18}O_P$ signature. Especially DOM, containing up to 45% O by weight has been shown
to persist until the precipitation of $Ag_3PO_4$ (McLaughlin et al., 2004). In literature, remaining O-
bearing compounds potentially being incorporated into the $Ag_3PO_4$ seems to be of bigger concern
(Davies et al., 2014; Goldhammer et al., 2011; Gruau et al., 2005; McLaughlin et al., 2004; Tamburini





329 et al., 2010) than the probability of acid hydrolysis during brucite dissolution (Jaisi & Blake, 2014;
330 Paytan & McLaughlin, 2011).

331 An alternative to the MagIC protocol is the quantitative removal of $P_i$ by co-precipitation with
332 Fe-oxides (Longinelli et al., 1976) which has been proven successful for freshwater samples (Gruau
333 et al., 2005; Neidhardt et al., 2018). Fe-oxide co-precipitation is initiated by addition of 0.1 M $FeSO_4$
334 accompanied with aeration of the sample at a pH of 8.5±0.1 (optimal for Fe-oxide precipitation; Gruau
335 et al., 2005). Neidhardt et al. (2018) found that it is not necessary to add $FeSO_4$ if the initial dissolved
336 $Fe^{2+}$ concentration in the solution samples are high (>1 mg $Fe^{2+}$/L). It still needs to be investigated
337 which which of the two approaches for quantitative $P_i$ removal is preferable.

### 2.4. Purification and silver phosphate precipitation

339 The phosphate purification protocol presented in the precent study is based on the method
340 published by Tamburini et al. (2010). The advantage of Tamburini et al. (2010)'s protocol is that it was
341 developed with the specific goal of minimizing the effect of organic matter on $\delta^{18}O_P$. Tamburini et al.
342 (2010)'s purification steps of sequential precipitation and recrystallization were adapted from
343 Kolodny et al. (1983) and modified by Liang & Blake (2006). Briefly, $P_i$ is first precipitated as
344 ammonium phospho-molybdate (APM), and then recrystallized as magnesium ammonium
345 phosphate (MAP). This is combined with a subsequent cation resin treatment followed by elimination
346 of chloride. The purification protocol is presented below.

### Step VII. Ammonium phospho-molybdate (APM) precipitation

348 During the first step of the purification protocol, $P_i$ is scavenged from the acidic dissolved brucite
349 solution by precipitation of APM crystals. This enables the separation and removal of ions and
350 contaminants that are soluble at low pH (Joshi et al., 2018). The APM precipitation procedure is as
351 follows:

352 *a)* Initially, transfer the sample solution (i.e. the dissolved brucite) to an Erlenmeyer flask of
353 suitable volume (sample and reactants combined volume) and place the flask in a 50 °C warm
354 water bath shaker or on a magnetic stirrer with heating set to 50 °C. *b)* If the solution is taken
355 directly from the refrigerator, wait until the sample is close to room temperature before
356 continuing. *c)* Then add 25 mL 35% ammonium nitrate reagent, and then slowly add 40 mL of
357 the 10% $NH_4$-molybdate solution. *d)* Adjust the final pH to ca. 1 using 1 M $H_2SO_4$, (use indicator
358 pH test strips). Normally around 1 mL is enough; thereby the volume of the sample is not
359 affected too much. *Note* that if the supernatant turns transparent bright yellow (Figure 3a) this
360 is an indication that optimal precipitation condition with respect to APM crystals are obtained.
361 When this color changes to milky yellow, it indicates that APM crystals are forming (Figure 3b).
362 If no APM crystals have started to precipitate from the heated solution (> 25 °C) after around 15
363 min, supersaturated conditions with respect to APM crystals are likely not obtained or pH is not
364 correctly adjusted. First check the pH and adjust if necessary. If still no APM crystals precipitate,
365 add stepwise more 35% ammonium nitrate and 10% $NH_4$-molybdate solution in the same ratio
366 as before (2.5:4) until signs of crystal precipitation. *e)* Leave the solution in the 50 °C warm water
367 bath and shake gently overnight to ensure complete APM precipitation.

369 *We experienced* that if the supernatants were slightly alkaline after Step VIIc it became bright green
370 (Figure 3c) and no APM started to precipitate. When adjusting the pH to 1 the supernatant turned to
371 a transparent bright yellow color (Figure 3a) and APM crystal immediately began to form. The
372 slightly alkaline condition could have affected the dissolution of the brucite flocs, since brucite
373 dissolve at acidic conditions flocs. We also experienced that if the brucite flocs had not been acidified
374 to pH 1 during the dissolution step (Step IV) and/or the additional MagIC step (Step V) was not
375 conducted, the crystals precipitating in this purification step were white and the supernatant
376 transparent (Figure 3d). Furthermore, we were not able to accomplish a final precipitation of $Ag_3PO_4$
377 when we tried to proceed with these white crystals. Accordingly, we suggest that the color of the

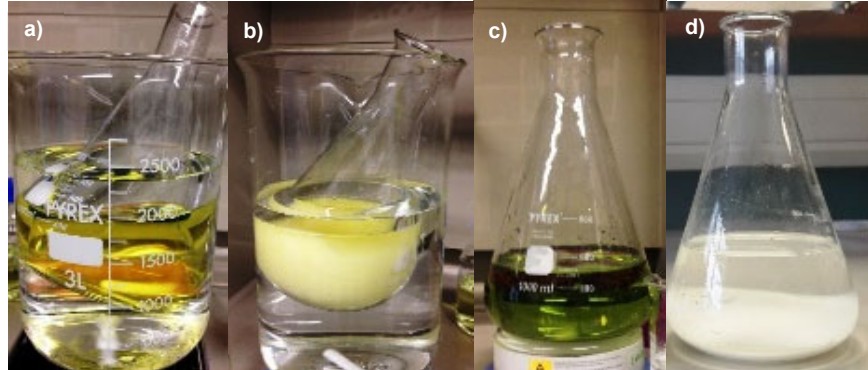

**Figure 3.** Color of the supernatant and the precipitate in Step VII when (a) optimal precipitation condition with respect to APM crystals are obtained, (b) APM crystals are forming, (c) alkaline condition which impede APM precipitation and (d) with an unidentified precipitate resulting from incorrect execution of the MagIC protocol (section 2.3).

supernatant and the precipitate can be used as an indicator for (*i*) optimal pH conditions for APM
precipitation and (*ii*) whether it is worthwhile to continue. The adjustment of the pH and an
introduction of additional MagIC steps were performed simultaneously in the present study. No
examination of whether both actions are equally important has been reported nor tested in the
present study.

**Step VIII. APM dissolution**
The $P_i$ is released from APM by dissolution of the crystals in an alkaline solution prior to an additional
purification step. Conduct the step as follows:
*a)* Start by separating the yellow APM crystals from the supernatant by vacuum filtration upon
a 0.2 μm cellulose acetate filter and discard the supernatant. The filtration time can take several
hours and more than one filter may be necessary. APM crystals from different samples may
differ slightly from each other in color and size (Figure 4). *b)* Wash the crystals thoroughly with
a 5% ammonium nitrate solution to rinse off contaminants. The more, the better (>200 mL). *c)*
Transfer the filter containing the APM crystals to a 100 mL Erlenmeyer flask and place the flask

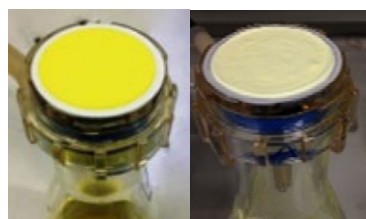

**Figure 4**. Vacuum filtered and 5% ammonium nitrate washed ammonium phospho-molybdate crystals (APM) from two different samples. The APM crystals differ in color and size.

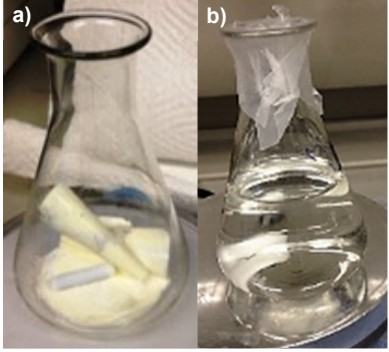

**Figure 5.** a) Ammonium phosphomolybdate crystals (APM) on 0.2 μm cellulose acetate filters. b) Dissolved APM crystal in a NH₄-citrate solution resulting in a transparent solution.

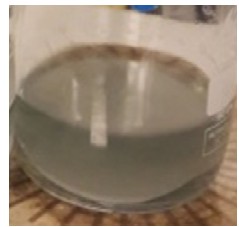

**Figure 6.** Greenish discoloration of the dissolved ammonium phosphor-molybdate crystals.





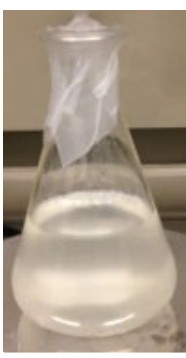

**Figure 7.** Precipitated magnesium
ammonium phosphate crystals .

on a magnetic stirrer. *d)* Dissolve the APM crystals in a minimum amount of $NH_4$-citrate solution
(15-50 mL; volume depends on the quantity of formed APM crystals). Start by adding 10 mL and
then add 5 mL aliquots. Work under a chemical fume hood. *e)* Gently swirl the solution while
the crystals are dissolving and wait until the solution becomes transparent (Figure 5), which may
take up to 15-20 minutes. Then remove and discard the filter(s). *Note* that $Mg^{2+}$ ions could
interfere with dissolution of the APM crystals leading to some crystals not dissolving. In
addition, silicates may have formed in the former steps. These are not dissolvable in the $NH_4$-
citrate solution. Accordingly, some particulate compounds might be left in the solution after it
turns transparent. If so, filter again using a 0.2 μm cellulose nitrate filter and discard the filter.
*We experienced* that the dissolved APM solution at times had a greenish discoloration, maybe due
to precipitation formation of silicate molybdate complexes (Figure 6). This could possible indicate
that silicate molybdate complexes have formed. We tried to continue the protocol with these samples
which still resulted in $Ag_3PO_4$ crystal precipitation in the last step.
**Step IX. Magnesium ammonium phosphate (MAP) precipitation**
In this step $P_i$ is further purified by precipitating MAP crystals under alkaline conditions, thus
enabling the removal of ions and contaminants that are soluble at high pH. The MAP precipitation
procedure is as follows:
*a)* Initially add 25 mL Mg-reagent to the 100 mL Erlenmeyer flask, containing the dissolved
APM solution, while stirring. *b)* Then slowly add about 7 mL of the 1:1 ammonia solution. *c)*
Check pH. If pH < 8 carefully add more of the 1:1 ammonia solution until the solution acquires
pH 8-9 which is the optimum pH for MAP precipitation. MAP crystals should start to
precipitate immediately, turning the solution whitish opaque (Figure 7). *d)* Cover the
Erlenmeyer flask with parafilm and make mm-size holes for venting. Leave the solution
overnight on the magnetic stirrer.
*We experienced* that it was necessary to add a bit more Mg-reagent to some of the samples, after
adjusting pH to 8-9, in order to achieve supersaturation with respect to the MAP crystals. This was
true for the samples were >20 mL of $NH_4$-citrate solution had been used to dissolved the APM
crystals.
**Step X. MAP dissolution**
$P_i$ is released by dissolving the MAP crystals in a minimum amount of $HNO_3$.
*a)* Separate the white MAP crystals from the supernatant by vacuum filtration upon a 0.2 μm
cellulose nitrate filter and discard the supernatant. The MAP crystals are small and may be quite





hard to see on the filter by eye, see Figure 8. *b)* Wash the crystals thoroughly with 1:20 ammonia
solution (>200 mL) to get rid of excess chloride and other contaminants. *Note* that this is of
extreme importance as remaining $Cl^-$ from the Mg solution (i.e. MgCl and HCl) will cause co-
precipitation of AgCl during the final precipitation of $Ag_3PO_4$. *c)* Transfer the filter to a 50 mL
centrifuge tube (with lid) and dissolve the MAP crystals in a minimum amount of 0.5 M $HNO_3$
(5-10 mL) by shaking the sample. *d)* Leave the filter in contact with the acid for at least 15-20
minutes to ensure that the MAP has dissolved. *Note* that it is difficult to assess when the crystal
have fully dissolved, since the filters and the crystals are both white.

### Step XI. Cation removal

The presence of cations (primarily $Na^+$ and multivalent cations such as $Mg^{2+}$) interferes with the
precipitation of $Ag_3PO_4$. Thus a prior cation removal step is a crucial prerequisite for the subsequent
successful precipitation of purified $Ag_3PO_4$ (Firsching, 1961). Cations can be scavenged by a proton-
charged cation resin, releasing $H^+$ to solution, which subsequently reacts with $HCO_3^-$ (if present),
forming $H_2O$ and $CO_2$ (Colman, 2002). The purification step is as follows:
*a)* Convert the new cation exchange resin AG50WX8 to an $H^+$ form by reacting the resin with 7
M $HNO_3$ overnight, on a horizontal shaker. A 7 M $HNO_3$ volume equivalent to 1.5 times the
resin volume is recommended. *b)* The following day discard the $HNO_3$ and rinse the resin
thoroughly by mixing it with 1 L DD-$H_2O$ to bring it close to neutrality (>5). *c)* Filtrate the
mixture on a 0.45 µm polycarbonate filter and discard the water. It might take up to several
repetitions before a neutral pH is obtained. *d)* Add 6 mL of the obtained cation resin slurry to
the sample solution. Seal the sample with a lid or parafilm and place the sample on a shaker
overnight. *e)* The next day, filter the sample using a 0.2 µm polycarbonate filter and rinse the
cation resin with 1-2 mL DD-$H_2O$. *f)* Collect the resin and recondition it in 1 M $HNO_3$. The resin
can be re-used.
*Method disagreement* regarding the preparation of the cation resin. Goldhammer et al. (2011)
experienced a reddish discoloration of the sample when using resin prepared the previous day.
Subsequently they were unable to properly precipitate $Ag_3PO_4$. By preparing the cation resin within
30 min of its use they avoided this problem. They did not resolve the cause of this complication. We
experienced that the samples acquired a milky white color once the resin was added, if the resin was
prepared two days before its use (our resin was left in DD-$H_2O$ overnight). The whitish coloration
was avoided when using the resin the same day as it was washed in DD-$H_2O$. It was not possible to
properly precipitate $Ag_3PO_4$ when using samples where the milky white color had occurred. Thus,
we agree with Goldhammer et al. (2011)'s statement, that proper handling and rinsing of the resin
before every application is crucial to the successful precipitation of $Ag_3PO_4$.

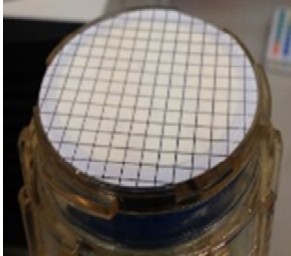

**Figure 8.** Vacuum filtered and 1:20 ammonia solution washed magnesium ammonium phosphate crystals.

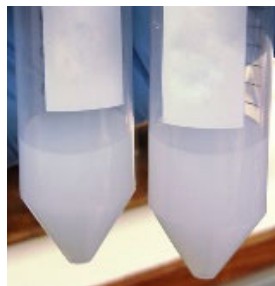

**Figure 9.** Precipitated AgCl crystals after adding
AgNO₃ to the sample solution.

Joshi et al. (2018) adjusted the pH of the dissolved MAP solution to neutral (pH 6-8) prior to the
cation removal.

**Step XII. Elimination of Cl⁻**

Removal of Cl⁻ ions is extremely important, as Cl⁻ otherwise may react with the Ag⁺ in the final
precipitation of AgPO₄, forming AgCl which have been observed to be rimmed by silver oxide
precipitates (Colman, 2002). Precipitation of AgCl hence both interfere with the Ag₃PO₄ precipitation
(McLaughlin et al., 2004) and introduce non-phosphate oxygen to the sample (Colman, 2002).
Chloride can be quantitatively removed by adding AgNO₃ crystals to the sample when the pH is
acidic, causing AgCl precipitation (Figure 9) prior to the Ag₃PO₄ precipitation step. The low pH in
the sample (<1) impede co-precipitation of Ag₃PO₄, and hence no P$_i$ is lost during this step. The
purification step is as follows:
*a)* Transfer the filtrated sample solution to a small container with a small opening (e.g. 50 mL
centrifuged tube) *b)* Add a few AgNO₃ crystals to the sample solution. If the sample turns
whitish opaque AgCl has precipitated (Figure 9). *d)* Wait at least 5 minutes and re-filter, the same
filter used in Step XI can be re-used.
After this purification step, the initial freshwater sample with a volume up to 50 L has been reduced
to about 10 mL highly concentered homogeneous P$_i$ solution stripped of potential contaminants. The
sample is now ready for the final Ag₃PO₄ precipitation.

**Step XIII. Silver phosphate (Ag₃PO₄) precipitation**

Precipitation of insoluble silver salts, such as Ag₃PO₄, can be conducted by volatilization of ammonia
(Firsching, 1961). This allows a 'slow' recrystallization which facilitates the growth of large and
easier-to-handle Ag₃PO₄ crystals for oxygen isotope analysis by IRMS within a few days (Firsching,
1961; Goldhammer et al., 2011). The method utilizes that Ag₃PO₄ precipitates in the solution at a pH
around 7±0.5 when free Ag⁺ and P$_i$ are present. Thus the pH conditions and a high Ag⁺:P$_i$ ratio is of
extreme importance to ensure complete precipitation of Ag₃PO₄. The 'slow' Ag₃PO₄ precipitation
procedure is as follows:
*a)* Initially add the Ag-ammine solution to the sample solution, in a Ag:P$_i$ ratio of approximately
10:1 (Colman, 2002). The sample solution turns briefly white (at pH 7), and then transparent (at
pH>7) once the alkaline Ag-ammine solution has been added. *b)* The sample container is then
placed in an oven at 50 °C. Yellow Ag₃PO₄ crystals start to precipitate after a few hours as the
amine starts to vaporize and the Ag⁺ is released (Firsching, 1961). Complete precipitation of the
crystals takes up to two days. *Note* that it impotent to repeatedly add DD-H₂O to the solution to
keep the volume as constant as possible. If left unattended (e.g. for one or several days) all the
H₂O may evaporate, which results in uncontrolled precipitation of salts. This is still fine, as the





salts will be dissolved when adding DD-H₂O, as they are mostly nitrate-based. If this happen it
is vital to wash the Ag₃PO₄ crystals easterly well with DD-H₂O. The small diameter of the tube
and the low temperature of the oven, impedes the evaporation ammonia, and thus enables a
slow crystallization process (Colman, 2002). *c)* After 1 to 2 days, if no yellow Ag₃PO₄ crystals
have precipitated, check the pH of the solution. If the pH of the solution differs from pH 7
(optimal pH for Ag₃PO₄ precipitation conditions; Firsching, 1961) adjust the pH by adding either
HNO₃ or NH₄OH. *Note* that under no circumstances should HCl or NaOH be used to adjust the
pH as Cl⁻ and Na⁺ would interfere with the crystallization of Ag₃PO₄. *d)* When crystals have
formed, vacuum filter them upon a 0.2 μm polycarbonate filter and discard the supernatant.
Other filters tend to 'trap' the Ag₃PO₄ crystals on their surface. *Note* that Ag₃PO₄ crystals may
form on the side of the tube, hence make sure to carefully detach these and transfer them to the
filter as well. *e)* Wash the crystals extremely thoroughly with DD-H₂O to get rid of residual O-
bearing compounds, as they interfere with the oxygen isotope analysis (cf. Section 2.3.1). *f)* Place
the filter on a Petri dish and cover it to prevent contamination and loss of crystals. Dry the filter
at 50 °C for at least 1 day. *g)* An extra elimination of residual organic matter might be necessary
by introducing a final washing of the Ag₃PO₄ precipitate with hydrogen peroxide to eliminate
residual organic matter by oxidation (Tamburini et al., 2010). *Note* that Crowson et al. (1991)
found that contaminated silver phosphate crystals were generally dark brown to greenish brown
in color and cohesive. We did not experience this discoloration but the crystal became dark
under light, probably do to the photo-oxidation of silver (McLaughlin et al., 2004; Tamburini et
al., 2010). *h)* If needed, the filter containing the Ag₃PO₄ crystals can be stored in a desiccator.
*Method disagreement* regarding the Ag₃PO₄ precipitation rate. The final precipitation of Ag₃PO₄ can be
accomplished by either a 'slow' (Goldhammer et al., 2011; Tamburini et al., 2010) or 'fast' (Dettman
et al., 2001; McLaughlin et al., 2004) precipitation method. In contrast to the 'slow' method presented
in the present protocol, 'fast' AgNO₃ precipitation is achieved by first altering the solution pH to 7 by
adding NH₄OH, NH₄NO₃ and nitric acid (HNO₃). Then follows the addition of AgNO₃ crystals
dissolved in DD-H₂O, which initiate a rapid precipitation of Ag₃PO₄ within a few minutes
(McLaughlin et al., 2004).
Dettman et al. (2001) compared the isotopic composition of the Ag₃PO₄ generated by the two
different methods and found the resulting δ¹⁸O$_P$ values to be within expected interlaboratory
variation. Tamburini et al. (2010) suggest, however, using the 'slow' precipitation method as an
additional measure to minimize the disturbance by organic matter as suggested by Colman (2002).

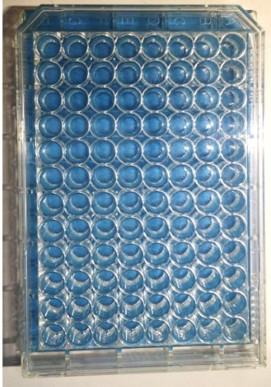

**Figure 10.** A microplate, used for transportation of
timble capsules containing Ag₃PO₄ crystals.





**Step XIV. Ag₃PO₄ crystal preparation prior to isotope ratio mass spectrometry (IRMS)**

Once the Ag₃PO₄ crystals have been precipitated, dried and stored in plastic vials they are prepared for isotope ratio mass spectrometry (IRMS) analysis. Such sample preparation involves:

*a)* Weighing of ~300 μg of Ag₃PO₄ crystals in triplicate from the plastic storage vial into silver timble capsules. *b)* After weighing the Ag₃PO₄ and recording the weight, add a small amount (few grains) of black carbon (no need to weigh this) to each sample. *c)* Close the capsules tight by using tweezers but absolutely do not touch them with the fingers. *d)* Place the capsules in a microplate with holes (Figure 10). Once all the samples have been weighed off, seal the plate. To do this, cover the plate with parafilm, close it with the cover lid, and then fix the lid with tape.

The Ag₃PO₄ crystal are now ready to be shipped for $\delta^{18}O_P$ analysis.

*Evaluation of phosphate purification and silver phosphate precipitation*

The effectiveness of the different purification steps, in producing adequately pure Ag₃PO₄, is difficult to evaluate during the execution. Oxygen contamination can not be checked for until $\delta^{18}O_P$ has been analyzed. The oxygen yield of the sample is compared to that of the pure Ag₃PO₄ used as standard (Tamburini et al., 2010). Therefore, it is important to know and pay attention to the characteristics of the precipitated crystals in each step (e.g. correct crystal color) and evaluate whether the specific purpose of the step has been obtained (e.g. whether crystals are formed or completely dissolved).

## 3. Final remarks

In general, it is important to keep in mind, that the amount of added reactants and chemicals can vary from water sample to water sample and in many instances it depends on yield volume or quantity from the prior step. Hence, only minimum and indicative quantities are stated in the present protocol.

As stated in the introduction, a variety of approaches have been attempted to address the problem regarding $\delta^{18}O_P$ contamination resulting from O-bearing compounds others than $P_i$, and from $Na^+$, $Cl^-$ and multivalent cations. Bearing in mind that many methods have been tested on various water matrix compositions, their effectiveness may not be reproducible for all water samples matrices. In order to achieve progress in developing and applying $\delta^{18}O_P$ to trace P sources and cycling in freshwater ecosystems, a better understanding of the different methods' reliance on different water matrices is crucial. This does not only apply to the purification steps but applies for all sections presented in the present study.

In general, studies which have used $\delta^{18}O_P$ as a tracer emphasize the importance of additional research and knowledge regarding $\delta^{18}O_P$ data for various potential phosphate sources especially for freshwater systems (Elsbury et al., 2009; Granger et al., 2017; Tamburini et al., 2014; Young et al., 2009). The detailed protocol provided in this study will hopefully contribute to enable a broader use of $\delta^{18}O_P$ signatures as such a tracing tool.

**Competing interests** The authors declare to have no competing interests.

**Acknowledgements** The authors thank Jörg Lewandowski (IGB, Berlin) for helpful comments to an early version of the manuscript. This study was funded by the Geocenter Denmark-grant 6-2015.



## Appendix A: Description of the preparation of all used reagents

Reagents used in Section 2.3:

- **3 M Mg-brine:** Dissolve 610 g $MgCl_2 \cdot 6H_2O$ (hexahydrate; MW 203.3 g/mol) in deionized distilled water ($DD-H_2O$) to a total volume of 1 L. After the salt has dissolved, filter the brine on a GF/F filter. The solution can be stored indefinitely.
- **1 M NaOH:** Dissolve 40 g NaOH pellets in deionized distilled water ($DD-H_2O$) to a total volume of 1 L. The solution can be stored indefinitely.
- **1 M $HNO_3$:** Add 66 mL of concentrated $HNO_3$ to 934 mL of $DD-H_2O$. The solution can be stored indefinitely.

Reagents used in Section 2.4:

- **35% ammonium nitrate reagent:** Dissolve 538.5 g ammonium nitrate salt (MW 80.052 g/mol) in 1000 mL DD- $H_2O$. Stir well to dissolve the salt completely. The solution can be stored.
- **5% ammonium nitrate reagent:** Dissolve 105.5 g ammonium nitrate salt in 2000 mL DD- $H_2O$. Stir well to dissolve the salt completely. The solution can be stored.
- **10% $NH_4$-molybdate solution:** This solution has to be prepared freshly by dissolving 53.3 g of ammonium molybdate salt (tetrahydrate form: 1235.86 g/mol) in 480 mL of DD- $H_2O$ (enough for approximately 12 samples). The solution CANNOT be stored.
- **Ammonium-citrate solution:** Add 300 mL of DD- $H_2O$ and 140 mL of concentrated $NH_4OH$ to 10 g of citric acid while working under a chemical fume hood. The solution is stable at room temperature and can be stored.
- **Mg-reagent:** Dissolve 50 g of $MgCl_2$ (hexa-hydrate salt, MW 203.3 g/mol) and 100 g of $NH_4Cl$ (MW 53.49 g/mol) in 500 mL $DD-H_2O$. Subsequently acidify the mixture to pH 1 with concentrated HCl. Finally, adjust the volume to 1 L with $DD-H_2O$. The solution is stable indefinitely and can thus be stored.
- **1:1 and 1:20 ammonia solutions:** Measure in a volumetric cylinder concentrated $NH_4OH$ (50 mL for the 1:1 and 100 mL for the 1:20). Pour into an appropriate glass bottle and dilute with $DD-H_2O$ (50 mL for the 1:1 and 1900 mL for the 1:20). The solution can be stored.
- **0.5 N $HNO_3$ solution:** Add 33 mL of concentrated $HNO_3$ to 967 mL of $DD-H_2O$. The solution can be stored.
- **Ag-ammine solution:** Dissolve 10.2 g of $AgNO_3$ salt (MW 169.87 g/mol) and 9.6 g of $NH_4NO_3$ in 81.5 mL of $DD-H_2O$. Subsequently add 18.5 mL of concentrated $NH_4OH$. The solution can be stored in the dark in an amber bottle.

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
