# Peer review of "Analysis of oxygen isotopes of inorganic phosphate (δ18Op) in freshwater: A detailed method description"

_Hydrology and Earth System Sciences, 2019_

## Referee Comment (RC1) · Anonymous Referee #1 · 15 Nov 2019

Analysis of oxygen isotopes of inorganic phosphate ($\delta18$Op) in freshwater: A detailed method description by Nisbeth et al.

A) General comments

The present manuscript submitted by Niesbeth et al. is supposed to be published as a technical note that provides a detailed description for the analysis of oxygen isotope ratios in inorganic phosphate obtained from freshwater samples. Such a detailed step-by-step guide is generally of great interest to the growing number of scientists using this approach to investigate sources and cycling of phosphorus in the environment. I have to admit this is the first time I review a technical note, but thanks to extensive lab

experience, I have a good impression of what a sound technical note should look like.

What I expect from a technical note is a clear, straight-forward, user-friendly step-by-step description of the method and its respective steps. Having said this, we are already at the major issue of this manuscript. In its present form, the manuscript is a) too long (technical notes in HESS should be "a few pages only"), and b) it carries characteristics that are typical for other manuscript types. For example, it appears to be more of a review then a technical note at some points, which include citations and lengthy discussions. While this approach is fine for a classical review paper, it does not belong into a technical note. If, for example, your technical method is not suitable for a certain type of water sample, so just say it and do not attempt to come up with lengthy discussions of why and how it could work (for example, see lines 300-302 or 312-315). In sum, the two general question the authors need to clarify is how should the manuscript should look like as a sound technical note, and what needs to be done to get it there.

-Getting the manuscript into the right form also involves a substantial shortening of the present text. By removing text, there is also the question how to keep the aspect of novelty and not just repeat what has been previously published in other original papers (e.g., by Tamburini et al.) or reviews (e.g., by Davies et al. 2014), because this would be just a repetition of what has already been published. I therefore suggest to use your "own experience" additions (i.e., the only truly new information provided to the reader) in order to provide a clear optimized method description. Finally, the revised manuscript would be entitled something like "Analysis of oxygen isotopes of inorganic phosphate ($\delta$18OPO4) in freshwater: Detailed description of an optimized method" or something similar in this direction.

-What I also miss is a clear recommendation regarding quality control. How to assure your $\delta$ 18PO4 signal does not change during the numerous sample processing steps? As already laid out by the authors, there migtht be pronounced issues with high organic waters, bearing the risk that organic P becomes hydrolyzed to PO4, thereby altering

the original $\delta$ 18PO4 signal.

-I further suggest to include a schematic overview of all the processes involved, which would also act as a sort of graphical guideline (see for example Davies et al. 2014, Figure 4 for such an example).

-To conclude, the manuscript in its present form requires substantial revision to meet the criteria of a technical report. Considering the value of such an optimized method description for the growing numbers of researchers working on aquatic P cycling, I would like to encourage the authors to submit a revised version of their manuscript. Given the general and specific comments (see below), I end up with the recommendation "reject with suggestion of resubmission"; however, it appears that HESS does not provide this recommendation option for reviewers, so I leave the decision to handling editor if it is going to be a "major revision" or "rejection".

B) Specific comments

-1.Introduction: We need to ask ourselves here the following question - do we really need a lengthy introduction regarding the application of $\delta$18OPO4 analysis in a technical note? I would strongly recommend to condense the entire 2-page long paragraph into a short paragraph of 3-5 sentences that refer to the common literature.

-In accordance with Coplen's 2011 "Guidelines and recommended terms for expression of stable‐isotope‐ratio and gas‐ratio measurement results" (Rap. Comm. Mass Spectr.) , I would recommend the consistent use of the term $\delta$18OPO4throughout the text and avoid Pi and other non-conventional terms.

-Keep consistency regarding chemical concentrations; there are molar concentrations but also mg/L in the text, this should be consistent.

-Chapter 2.2, Step I: I have great doubt that the described procedure will be suitable to prevent co-precipitation of dissolved PO4 by Fe-oxyhydroxides if you have high Fe2+ concentrations in your water samples. During pumping, subsequent storage and transportation, it is impossible to avoid diffusion of $O_2$ through pumping hose and plastic materials. This will in turn quickly react (within minutes) with the $Fe^{2+}$ and form Fe-oxides, which in turn coprecipitate dissolved $PO_4$ from solution. Do the authors have prove for a successful application of their suggestion? If not I suggest to go for other ways to isolate the dissolved $PO_4$ from solution. This also brings me to the question if you really need such a lengthy sampling description in general; but this depends on where you want to focus your technical note.

-Line 188: This also applies to all previous steps, which means samples need to be processed immediately after sampling to avoid potential microbial alterations.

C) Purely technical corrections at the very end ("technical corrections": typing errors, etc.). line 248: There is no such section

---

## Referee Comment (RC2) · Anonymous Referee #2 · 21 Nov 2019

**1. General comments**

This manuscript synthesises previously published methodological work alongside a significant body of observation and experience from laboratory processing of samples for $\delta18OP$ analysis. The aims of this contribution, i.e. to provide a methodological baseline to inform researchers who are new to the field of $\delta18OP$ and to promote increased standardisation of methodological protocols across $\delta18OP$ research, are necessary and the authors should be congratulated for focussing on these timely issues.

In general, for the purposes of a technical note, the manuscript does appear to be relatively long. This is largely due to the inclusion of previously published work, pre-

dominantly based on the methodology of Tamburini et al. (2010). I do recognise that valuable and new insights into this methodology are provided by the authors, based on their experiences of processing samples in the laboratory. However, I feel that the manuscript could be shortened in places through clearer signposting to details already published elsewhere (particularly in section 2.4), thereby re-focussing the manuscript on the body of new observations and experiences provided by the authors.

2. Specific comments

- Lines 27-30 – I would avoid placing such a strong focus on sole phosphorus (P) limitation of freshwaters here. For example, the role of nitrogen (N) limitation or N/P co-limitation in freshwaters is of growing interest. The potential to apply $\delta$18OP analyses to resolve questions of P, of N or of N/P co-limitation is of particular interest within the freshwater community.

- Lines 36-39 – the $\delta$18OP community in freshwaters was initially strongly attracted to $\delta$18OP as a potential tracer of P source, with the potential to inform new source apportionment models. However, the available evidence increasingly indicates that $\delta$18OP rarely acts as a conservative tracer of P source, certainly over larger spatial and temporal scales in catchments (as the authors note on lines 48-49). I would argue that there is far more potential power in using $\delta$18OP to understand processes controlling P cycling in freshwaters, rather than to focus too strongly on questions of source apportionment.

- Lines 40-41 – orthophosphate is indeed the primary form of P involved in transport across cytoplasmic membranes prior to intracellular metabolism. However, this does not mean that orthophosphate is the primary form of P cycled in ecosystems. For example, hydrolysis of organic compounds containing P, whether to meet metabolic demand for P or for the purposes of dephosphorylation prior to carbon (C) utilisation, may be an extremely important part of the P cycle within certain freshwater ecosystems. Processes other than uptake of orthophosphate into the intracellular environment are

also associated with isotopic fractionation/isotope effects and may therefore be probed through $\delta18OP$ analyses.

- Line 49-50 – temperature-dependent equilibrium fractionation between intracellular fluid-oxygen and phosphate-oxygen during intracellular metabolism of P is indeed an important fractionation that influences the $\delta18OP$ system. However, the authors should also highlight the fact that other processes that potentially influence the P cycle within freshwaters may lead to inheritance or kinetic isotope effects. For example, the hydrolysis of organic P compounds will involve some inheritance of oxygen atoms from the phosphate moiety in the source organic compound and some incorporation of water-oxygen atoms into the liberated phosphate molecule (accompanied by a kinetic fractionation). The balance between equilibrium and kinetic/inheritance isotope fractionation and effects will ultimately determine $\delta18OP$ within a freshwater sample. Unravelling these controls on $\delta18OP$ is currently one of the major challenges facing the $\delta18OP$ community, but one that offers the potential to gain new insights into the range of processes influencing P within freshwaters.

- Line 145 – the authors suggest that freshwater sample volumes of up to 50 L may be necessary to generate sufficient Ag3PO4 for analysis, assuming a P concentration of 0.4 $\mu$M. In my experience, P concentrations (as dissolved reactive P) are often « 0.4 $\mu$M, certainly in freshwaters in which P availability is particularly low and therefore highly likely to be limiting primary production. It is precisely these systems in which $\delta18OP$ may offer new insights into the P cycle. However, this will require researchers to deal with sample volumes that often exceed 100 L, which presents additional methodological challenges.

- Line 173 – I agree that it is impractical to filter 50-100 L of sample straight through 0.45 $\mu$m filter papers. In my experience, sequential filtration starting with filter pore sizes >0.45 $\mu$m is required to address this issue. The risk of processing samples that have not been filtered is: i) dissolution of particulate-bound inorganic P; and/or ii) acid hydrolysis of particulate organic P. Either may generate phosphate with the potential

to alter $\delta$18OP, compared to the true $\delta$18OP of the original sample. In my opinion, standardising filtration of freshwater samples as part of any future $\delta$18OP analytical protocol is important.

- Section 2.3 – the authors focus on the use of the MagIC protocol as the initial processing step for freshwater samples. However, my experience means that I have significant doubts about the feasibility and accuracy of using MagIC in this way for large-volume freshwater samples. This is particularly true of freshwaters in which C:P ratios are high and in which there are much higher concentrations of dissolved organic matter (DOM) compared to the marine samples in which McLaughlin et al. (2004) originally developed their $\delta$18OP methodology that included MagIC. The specific challenges facing MagIC in freshwater matrices are:

i) The formation of a precipitate that does not subsequently redissolve in 1 M HNO3 (Step IV in the current manuscript). We have observed this within a number of freshwater matrices. We have not identified the precipitate, but the lack of redissolution in 1 M HNO3 suggests it is not brucite. When this occurs, our experience is that Ag3PO4 cannot be generated from a sample.

ii) Brucite is not specific for the phosphate ion and our research suggests that other competing oxyanions, including nitrate and sulphate, may be co-precipitated (this is in contrast to the statement made by the authors on lines 180-183).

iii) Dissolved organic matter, including a range of organic P compounds, can also be co-precipitated with brucite. In contrast to the research referenced on lines 309-310 of the manuscript, other observations suggest that acid hydrolysis of organic P compounds co-precipitated with brucite may indeed occur (see Davies et al. (2014) Figure 5 for example). At a minimum, we cannot yet concluded that no co-precipitation + acid hydrolysis of organic P compounds occurs in freshwater samples when using the MagIC methodology. Further work is needed to establish whether this has the potential to introduce errors into $\delta$18OP analyses in freshwaters, as a result of phosphate

generated from the organic P compound that differs in isotopic composition compared to phosphate within the original sample.

In light of points i) to iii) above, the authors may want to incorporate recent research in their manuscript that has developed alternative methodologies for $\delta$18OP analysis, seeking to avoid these potential sources of error. These methodologies primarily involve initial treatment of freshwaters using anion exchange resin to isolate phosphate from contaminant sources of oxygen, whether in organic matter or in the form of other oxyanions. For example, see: Tcaci, M. et al. (2019) A New Technique to Determine the Phosphate Oxygen Isotope Composition of Freshwater Samples at Low Ambient Phosphate Concentration. Environmental Science and Technology 53: 10288-10294 and Gooddy, D.C . et al. (2015) Isotopic fingerprint for phosphorus in drinking water supplies. Environmental Science and Technology. 49: 9020-9028.

- Line 200 – our observations suggest that the brucite precipitate can begin to re-dissolve if left after adding NaOH, likely because solution pH begins to decrease and brucite becomes unstable. Researchers should be cautious about this and be prepared to add further NaOH to maintain solution pH >9-10 in order to prevent the brucite (and co-precipitated phosphate) from redissolving into solution.

- Line 274-276 – do the authors know what these additional insoluble contaminants are and are they sure that they do not contain P?

- Line 316-319 – I agree that the use of labelled and unlabelled reagents is a way to assess potential error due to hydrolysis of organic P compounds. However, are the authors suggesting that this should be incorporated as standard practice in all $\delta$18OP analyses? How feasible is this? If not feasible, then we need a protocol that we can be certain does not risk hydrolysis of organic P compounds. As I comment on above, I'm not convinced that this is the case with the use of MagIC for freshwater samples.

- Line 425-428 – Why is washing to remove Cl- so important here, given that Step XII eliminates excess Cl- prior to Ag3PO4 precipitation by adding AgNO3?

[Figure]

- Line 449-452 – similar to Goldhammer et al. (2011), our observations also suggest AG50WX8 resin may generate a pink colour in solution, despite resin preparation using HNO3 and DD-H2O followed by immediate use. However, in contrast to Goldhammer, we did not observe any adverse effect on Ag3PO4 precipitation, although we follow the precipitation stages in the McLaughlin et al. (2004) method, rather than the Tamburini et al. (2010) method.

- Line 504-505 – which O-bearing compounds do the authors expect to be removed through this washing step?

- Line 525 – prior to this step, I assume crystals need to be removed from the filter papers and added to plastic vials? What experience do the authors have with this process, for example are the Ag3PO4 crystals difficult to handle due to static electricity?

- Line 535 – one of the risks of a multi-stage purification/precipitation process is the loss of phosphate in solution or of P in a solid precipitate during processing, for example if MAP hasn't fully dissolved (lines 431-432). Do the authors have any feeling for how significant these 'losses' of P may be within their protocol? Would this ever lead to an insufficient mass of Ag3PO4 being generated for analysis, despite an apparently sufficient mass of P being present in the initial freshwater sample?

3. Technical corrections

Line 245 – depends on, rather than deviate? Line 270 – wording of '….and thy all utilizes a pH…' needs to be clarified. Line 277 – Filter, rather than filtrate? Check this throughout the manuscript. Line 339 – present, rather than precent? Line 343 – modified from, rather than modified by? Line 470 – filtered, rather than filtrated? Check throughout the manuscript. Line 490 – important, rather than impotent? Line 494 – not sue what is meant by easterly?

---

## Referee Comment (RC3) · Anonymous Referee #3 · 9 Dec 2019

This MS essentially reads as a long protocol on a method to process freshwater samples to recover sufficient phosphate to allow for isotopic analysis of 18O:16O ratios. While there are a number of published approaches to do so for soils, marine waters, vegetation and other P bearing compounds, the lack of an approach for freshwaters has challenged researchers interested in developing a better understanding of P sources and cycling in freshwater systems. While I find the MS a bit long and perhaps dry in terms of discussion, I note that many of the observations documented within are similar to experiences that myself and others have observed when trying various approaches. Importantly, commentary on what does not work within a given step is a valuable contribution, particularly when one considers time and financial costs to go from large sample

volumes to a few grains of silver phosphate. For those of us working on small budgets, such information is extremely important. In the interests of shortening the MS, perhaps the authors might consider a tabular format which links agreement/disagreement with prior literature support as well as known issues with P recovery? Specific comments Lines 40-52 – given that in freshwaters, P is most often limiting, some indicate of the range of P found in freshwaters might be useful for readers. Further, some discussion of the relatively likelihood of where 18O-PO4 could be expected to function as a tracer (i.e. low light, eutrophic) vs simply an indication of probably P-recycling pathways (high light, oliogotrophic) might also be useful.

Line 145-6 – we have used volumes of up to 200 L for low P lakewater samples as typical concentrations are $\sim$ 0.05 uM. 0.4 uM is very high for many freshwaters in the northern hemisphere unless significantly affected by human activity or high natural sources of P.

Line 173-75 While in principle this is impractical, we have used sequences of 1 um and 0.2 um polypropylene cartridge filters under low pressure to filter >150 L down to 0.2 um with little problem.

Lines 180-82 – we have found that NaOH loading (as % v:v) has a strong influence on recovery of DOP/DOM.

Step V – In our experience, working in high C:P waters, NaOH loadings of $\sim$ 1 % v:v are required to completely remove Pi from supernatant in presence of highly colored brucite floc. Furthermore, once we have reduced to a working volume (ca 250 mL) the flocs become more intensely colored rather than less intensely colored as described here. Further, as described in Step VI, after acid dissolution at this stage, the is a darkly colored precipitate that does not dissolve. While we have not determined the nature of this precipitate, loss on ignition when combusted after lyophilisation is near 80% suggesting some acid insoluble organics. We have used columns (similar to Coleman 2002) to "clean" this brownish color from samples with good success.

Lines 287-302 – It should be worth adding that significant work on MAGIC in freshwaters has been done with respect to quantitation of Pi (i.e. Anagnostu and Sherrell 2008) and has identified other interfering compounds worth mentioning.
* * *

---

## Referee Comment (RC4) · Anonymous Referee #4 · 11 Dec 2019

The manuscript by Nisbeth and colleagues describes a protocol to extract phosphate from freshwater samples for the analysis of phosphate oxygen isotope signatures ($\delta$18Op). There are several protocols for this extraction to date, all having been developed for different sample matrices (seawater, freshwater, soil solution, waste water et cetera). One aim of the paper is to critically review these protocols for their adaptability to freshwater samples (which would be very welcome), and another is to provide the reader with a detailed method description on how to convert freshwater phosphate to IRMS-grade silver phosphate (which would be even more welcome).

However, the authors have decided to interweave both parts (the method review and

the protocol), and therefore its usability as a concise method description is limited. It is also on the lengthy side for what I would expect from a technical note. Thus, it would strongly benefit from a better organisation; maybe a structure that separates the review part from the description of the (novel) method.

These formal issues aside, I am missing a robust, data-backed method validation. What have the authors undertaken to verify the integrity of the method? There is unfortunately no data on, for example, phosphate recovery in the individual steps, Ag3PO4 yields, $\delta$18Op conservation. I consider this absolutely mandatory for a method paper (as a HESS technical note or elsewhere), and I strongly suggest that the authors include such evidence in a thorough revision before further consideration.

Other suggestions:

Title. A little misleading, because the actual measurement of d18OP is not part of the paper (Analysis of oxygen isotopes of inorganic phosphate ($\delta$18Op) in freshwater: A detailed method description) – more apt would be something like "Purification of silver phosphate from freshwater samples for the determination of $\delta$18Op".

Line 199 and elsewhere. Mg-brine is probably not a common term, why not using MgCl2? That said, I would avoid adding chloride anyway because one will have to get rid of it before Ag3PO4 precipitation.

Quality of the figures. The photos appear rather blurry and the presentation of lab equipment/vials/glassware would benefit from better organization. Some photos may be redundant (e.g. Fig 10).
* * *

---

## Author Comment (AC1) · 17 Jan 2020

Anonymous Referee #1 Analysis of oxygen isotopes of inorganic phosphate (_18Op) in freshwater: A detailed method description by Nisbeth et al.

A) General comments The present manuscript submitted by Niesbeth et al. is supposed to be published as a technical note that provides a detailed description for the analysis of oxygen isotope ratios in inorganic phosphate obtained from freshwater samples. Such a detailed stepby-step guide is generally of great interest to the growing number of scientists usingthis approach to investigate sources and cycling of phosphorus in the environment. I have to admit this is the first time I review a technical note, but thanks to extensive lab experience, I have a good impression of what a sound technical note should look like. What I expect from a technical note is a clear, straight-forward, user-friendly stepby- step description of the method and its respective steps. Having said this, we are already at the major issue of this manuscript. In its present form, the manuscript is a) too long (technical notes in HESS should be "a few pages only"), and b) it carries characteristics that are typical for other manuscript types. For example, it appears to be more of a review then a technical note at some points, which include citations and lengthy discussions. While this approach is fine for a classical review paper, it does not belong into a technical note. If, for example, your technical method is not suitable for a certain type of water sample, so just say it and do not attempt to come up with lengthy discussions of why and how it could work (for example, see lines 300-302 or 312-315). In sum, the two general question the authors need to clarify is how should the manuscript should look like as a sound technical note, and what needs to be done to get it there.

Firstly, I would greatly wish to thank the reviewer for his insightful and helpful comments in relation to the need and compilation of such a technical note. We will endeavour to (a) cut down on the length of the technical note. (This request also comes from the two other reviewers). (b) We will endeavour to remove any deemed unnecessary further discussion as requested but just to clarify what we were trying to highlight here was the background and reasons for the steps, and it appears to us that including such background and reasons is accepted or even appreciated (e.g., R#3) by the other reviewers; although all reviewers note the length. Giving a reason as to the background of each step to some small degree is very important in our opinion as it helps the user estimate how their samples will behave and cut down on costly mistakes. Finally, we submitted this as a 'technical note' because we think it suits that category more than it would fit the (also more prestigious) 'original research' or 'review' categories, although

we also acknowledge that it is not the most typical example of a 'technical note'.

-Getting the manuscript into the right form also involves a substantial shortening of the present text. By removing text, there is also the question how to keep the aspect of novelty and not just repeat what has been previously published in other original papers (e.g., by Tamburini et al.) or reviews (e.g., by Davies et al. 2014), because this would be just a repetition of what has already been published. I therefore suggest to use your "own experience" additions (i.e., the only truly new information provided to the reader) in order to provide a clear optimized method description. Finally, the revised manuscript would be entitled something like "Analysis of oxygen isotopes of inorganic phosphate (_18OPO4) in freshwater: Detailed description of an optimized method" or something similar in this direction.

We can take on this helpful advice and attempt to include our "own experience" sections as we go through the document. We can also change the title to something more similar to what has been suggested. R#4 propose a somewhat different title, though. Further, we plan to conduct new experiments to add new, validating, data to the paper (to concur to other of the reviewers' valuable comments, cf. the next comment of R#1); when this is done and the paper is revised, the current title may be still valid, perhaps.

-What I also miss is a clear recommendation regarding quality control. How to assure your _ 18PO4 signal does not change during the numerous sample processing steps? As already laid out by the authors, there might be pronounced issues with high organic waters, bearing the risk that organic P becomes hydrolyzed to PO4, thereby altering the original _ 18PO4 signal.

The original description of the brucite step for the purification includes some discussion on the stripping out of organic matter (Colman thesis). There are two possible solutions: One is to use the suggested low molarity of NaOH and consequently low molarity of the acid (as suggested in Colman), or to monitor the possible DOP hydrolysis by using 18O labelled and unlabelled acids to hydrolyse the brucite floc (as we

do for the HCl extraction in soils, see my paper Tamburini 2010). The downside of this second approach is that you need duplicates of samples, thus, big amounts of water.

-I further suggest to include a schematic overview of all the processes involved, which would also act as a sort of graphical guideline (see for example Davies et al. 2014, Figure 4 for such an example).

We can accommodate this by constructing such a schematic as suggested.

-To conclude, the manuscript in its present form requires substantial revision to meet the criteria of a technical report. Considering the value of such an optimized method description for the growing numbers of researchers working on aquatic P cycling, I would like to encourage the authors to submit a revised version of their manuscript.

We appreciate that the reviewer acknowledges the subject and its timeliness. As authors we will endeavour to improve the document considerably if we are given the opportunity to correct and upgrade the manuscript based the good suggestions outlined in this review and the others.

Given the general and specific comments (see below), I end up with the recommendation "reject with suggestion of resubmission"; however, it appears that HESS does not provide this recommendation option for reviewers, so I leave the decision to handling editor if it is going to be a "major revision" or "rejection".

B) Specific comments -1.Introduction: We need to ask ourselves here the following question - do we really need a lengthy introduction regarding the application of _18OPO4 analysis in a technical note? I would strongly recommend to condense the entire 2-page long paragraph into a short paragraph of 3-5 sentences that refer to the common literature.

I agreed to some degree. However, in our opinion, we believe we need more of an introduction than 3-5 sentences in introducing this technical note. I think it would be more reasonable to edit the introduction down to approx. 1 page in order to do justice

relating to the motivations and importance in the difficulty of the method but also the positive motivation of the work.

-In accordance with Coplen's 2011 "Guidelines and recommended terms for expression of stableâËŸA ËĞ RisotopeâËŸA ËĞ Rratio and gasâËŸARËĞ ratio measurement results" (Rap. Comm. Mass Spectr.) , I would recommend the consistent use of the term _18OPO4throughout the text and avoid Pi and other non-conventional terms.

This is taken on-board and will be incorporated into the document.

-Keep consistency regarding chemical concentrations; there are molar concentrations but also mg/L in the text, this should be consistent.

We agree; this is a clear mistake and we will use consistent chemical concentrations in the revised manuscript.

-Chapter 2.2, Step I: I have great doubt that the described procedure will be suitable to prevent co-precipitation of dissolved $PO_4$ by Fe-oxyhydroxides if you have high $Fe^{2+}$ concentrations in your water samples. During pumping, subsequent storage and transportation, it is impossible to avoid diffusion of $O_2$ through pumping hose and plastic materials. This will in turn quickly react (within minutes) with the $Fe^{2+}$ and form Fe-oxides, which in turn co-precipitate dissolved $PO_4$ from solution. Do the authors have prove for a successful application of their suggestion? If not I suggest to go for other ways to isolate the dissolved $PO_4$ from solution. This also brings me to the question if you really need such a lengthy sampling description in general; but this depends on where you want to focus your technical note.

Well, we do not know to which extent freshwater from "normal" rivers have reduced conditions and contain a lot of $Fe^{2+}$. We agree that even if the precautions proposed in 2.2 are taken, still $PO_4$-iron-oxide co-precipitation may (will, to some degree) occur. In practice, however, the suggested steps in 2.2 are more precautious than what many field workers will have done so far. In the case we have such waters, we could co-
precipitate brucite and Fe-bound P. A step with a chelating or reducing agent could be added (either dithionite or Na-EDTA). Anyways – and this we will clarify in the revised manuscript's introduction – the paper's main objective is not to present a 'now finally completely validated 18Op-freshwater method'. Instead, the objective is to present a consensus for the current way to sample and isolate 18Op of freshwater samples (cf., the general comment of R#2).

Our wish for the scientific community is that the method description that the paper provides can be used as stepping stone for further development of the 18Op method. For example, the paper should clearly point out where validation exists (by reference) and where validation is still needed.

Nevertheless, we will shorten the text in 2.2; we admit it is too verbose.

-Line 188: This also applies to all previous steps, which means samples need to be processed immediately after sampling to avoid potential microbial alterations.

The main idea is that the water sample is treated right after sampling with the NaOH to precipitate brucite. In theory, brucite precipitate at high pH, a condition where most of microbial activity is prevented. In addition, whenever possible, the brucite sample should be stored in a cold room, if time is needed before centrifugation and further processing. Microbial alteration is minimized.

C) Purely technical corrections at the very end ("technical corrections": typing errors, etc.). line 248: There is no such section Interactive comment on Hydrol. Earth Syst. Sci. Discuss., https://doi.org/10.5194/hess-2019- 469, 2019. This is a typo and will be removed, there was no such section.

---

## Author Comment (AC2) · 17 Jan 2020

1. General comments This manuscript synthesises previously published methodological work alongside a significant body of observation and experience from laboratory processing of samples for_18OP analysis. The aims of this contribution, i.e. to provide a methodological baseline to inform researchers who are new to the field of _18OP

and to promote increased standardisation of methodological protocols across _18OP research, are necessary and the authors should be congratulated for focussing on these timely issues.

In general, for the purposes of a technical note, the manuscript does appear to be relatively long. This is largely due to the inclusion of previously published work, predominantly based on the methodology of Tamburini et al. (2010). I do recognise that valuable and new insights into this methodology are provided by the authors, based on their experiences of processing samples in the laboratory. However, I feel that the manuscript could be shortened in places through clearer signposting to details already published elsewhere (particularly in section 2.4), thereby re-focussing the manuscript on the body of new observations and experiences provided by the authors.

We can take this helpful advice on-board and endeavour to shorten parts of the methodology and take into account our own observations and experiences as we go through the steps. We still want to present the complete method, though, but will focus more on the relation to our own observations and experiences as we go through the steps.

2. Specific comments

- Lines 27-30 – I would avoid placing such a strong focus on sole phosphorus (P) limitation of freshwaters here. For example, the role of nitrogen (N) limitation or N/P colimitation in freshwaters is of growing interest. The potential to apply _18OP analyses to resolve questions of P, of N or of N/P co-limitation is of particular interest within them freshwater community.

Certainly, we can briefly mention the growing interest in the role of nitrogen (N) limitation and N/P colimitation in freshwaters in our introduction.

- Lines 36-39 – the _18OP community in freshwaters was initially strongly attracted to_18OP as a potential tracer of P source, with the potential to inform new source

apportionment models. However, the available evidence increasingly indicates that _18OP rarely acts as a conservative tracer of P source, certainly over larger spatial and temporal scales in catchments (as the authors note on lines 48-49). I would argue that there is far more potential power in using _18OP to understand processes controlling P cycling in freshwaters, rather than to focus too strongly on questions of source apportionment.

Certainly, as suggested we can highlight the potential power in using 18OPO4 to understand processes controlling P cycling in freshwaters, rather than focus too much on source apportionment. We will clarify this in the revised manuscript.

- Lines 40-41 – orthophosphate is indeed the primary form of P involved in transport across cytoplasmic membranes prior to intracellular metabolism. However, this does not mean that orthophosphate is the primary form of P cycled in ecosystems. For example, hydrolysis of organic compounds containing P, whether to meet metabolic demand for P or for the purposes of dephosphorylation prior to carbon (C) utilisation, may be an extremely important part of the P cycle within certain freshwater ecosystems. Processes other than uptake of orthophosphate into the intracellular environment are also associated with isotopic fractionation/isotope effects and may therefore be probed through _18OP analyses.

This is a very good point and will be included in our corrections/edits of the manuscript.

- Line 49-50 – temperature-dependent equilibrium fractionation between intracellular fluid-oxygen and phosphate-oxygen during intracellular metabolism of P is indeed an important fractionation that influences the _18OP system. However, the authors should also highlight the fact that other processes that potentially influence the P cycle within freshwaters may lead to inheritance or kinetic isotope effects. For example, the hydrolysis of organic P compounds will involve some inheritance of oxygen atoms from the phosphate moiety in the source organic compound and some incorporation of water oxygen atoms into the liberated phosphate molecule (accompanied by a kinetic

fractionation). The balance between equilibrium and kinetic/inheritance isotope fractionation and effects will ultimately determine _18OP within a freshwater sample. Unravelling these controls on _18OP is currently one of the major challenges facing the _18OP community, but one that offers the potential to gain new insights into the range of processes influencing P within freshwaters.

Certainly, we will include this very valuable point on the inheritance or kinetic isotope effects which potentially influence the P cycle in freshwaters.

- Line 145 – the authors suggest that freshwater sample volumes of up to 50 L may be necessary to generate sufficient Ag3PO4 for analysis, assuming a P concentration of 0.4 $\mu$M. In my experience, P concentrations (as dissolved reactive P) are often Âń 0.4 $\mu$M, certainly in freshwaters in which P availability is particularly low and therefore highly likely to be limiting primary production. It is precisely these systems in which _18OP may offer new insights into the P cycle. However, this will require researchers to deal with sample volumes that often exceed 100 L, which presents additional methodological challenges.

Again, this is a very valid and important point which will be included in the corrected manuscript. Recently Tcaci (2019) published an article giving a new procedure for treating large volumes of water is described.

- Line 173 – I agree that it is impractical to filter 50-100 L of sample straight through 0.45 $\mu$m filter papers. In my experience, sequential filtration starting with filter pore sizes >0.45 $\mu$m is required to address this issue. The risk of processing samples that have not been filtered is: i) dissolution of particulate-bound inorganic P; and/or ii) acid hydrolysis of particulate organic P. Either may generate phosphate with the potential to alter _18OP, compared to the true _18OP of the original sample. In my opinion, standardising filtration of freshwater samples as part of any future _18OP analytical protocol is important.

This is an important point which we can expand and make suggestions in our corrected manuscript. We have filtered samples through 100 micron plastic screens with good results but we will include the suggestion of a sequential filtering protocol if it is deemed necessary in waters with a lot of particulates. In ferrous waters, however, a lengthy filtration procedure (slow pumping velocity?) could cause more damage than good, depending on the effect of co-precipitation of PO4 with iron oxides. Clearly, the magnitude of the error introduced by allowing particulates into the high-volume sample required attention in future research. This we will point out.

- Section 2.3 – the authors focus on the use of the MagIC protocol as the initial processing step for freshwater samples. However, my experience means that I have significant doubts about the feasibility and accuracy of using MagIC in this way for large-volume freshwater samples. This is particularly true of freshwaters in which C:P ratios are high and in which there are much higher concentrations of dissolved organic matter (DOM) compared to the marine samples in which McLaughlin et al. (2004) originally developed their _18OP methodology that included MagIC. The specific challenges facing MagIC in freshwater matrices are:

i) The formation of a precipitate that does not subsequently redissolve in 1 M HNO3 (Step IV in the current manuscript). We have observed this within a number of freshwater matrices. We have not identified the precipitate, but the lack of redissolution in 1 M HNO3 suggests it is not brucite. When this occurs, our experience is that Ag3PO4 cannot be generated from a sample.

ii) Brucite is not specific for the phosphate ion and our research suggests that other competing oxyanions, including nitrate and sulphate, may be co-precipitated (this is in contrast to the statement made by the authors on lines 180-183).

iii) Dissolved organic matter, including a range of organic P compounds, can also be co-precipitated with brucite. In contrast to the research referenced on lines 309-310 of the manuscript, other observations suggest that acid hydrolysis of organic P compounds co-precipitated with brucite may indeed occur (see Davies et al. (2014) Figure

5 for example). At a minimum, we cannot yet concluded that no co-precipitation + acid hydrolysis of organic P compounds occurs in freshwater samples when using the MagIC methodology. Further work is needed to establish whether this has the potential to introduce errors into _18OP analyses in freshwaters, as a result of phosphate generated from the organic P compound that differs in isotopic composition compared to phosphate within the original sample.

In light of points i) to iii) above, the authors may want to incorporate recent research in their manuscript that has developed alternative methodologies for _18OP analysis, seeking to avoid these potential sources of error. These methodologies primarily involve initial treatment of freshwaters using anion exchange resin to isolate phosphate from contaminant sources of oxygen, whether in organic matter or in the form of other oxyanions. For example, see: Tcaci, M. et al. (2019) A New Technique to Determine the Phosphate Oxygen Isotope Composition of Freshwater Samples at Low Ambient Phosphate Concentration. Environmental Science and Technology 53: 10288-10294 and Gooddy, D.C . et al. (2015) Isotopic fingerprint for phosphorus in drinking water supplies. Environmental Science and Technology. 49: 9020-9028.

We can include the valuable new points made by the reviewer in lines 309-310. In relation to points (i) and (ii), we will refer to the Tcaci paper (2019) for large volumes and using of labelled and unlabelled acid to track the possible hydrolysis of DOP during dissolution of brucite.

- Line 200 – our observations suggest that the brucite precipitate can begin to re-dissolve if left after adding NaOH, likely because solution pH begins to decrease and brucite becomes unstable. Researchers should be cautious about this and be prepared to add further NaOH to maintain solution pH >9-10 in order to prevent the brucite (and co-precipitated phosphate) from redissolving into solution.

We can emphasis this point and include the additional information. We suggest to start the centrifugation and dissolution of the brucite not long after its precipitation This was

also a point by Colman. So, I would not spend too much time here, just say that brucite should not be left for hours sitting there.

- Line 274-276 – do the authors know what these additional insoluble contaminants are and are they sure that they do not contain P?

We are not sure these insoluble contaminants do not contain P. We will change the word 'contaminants' to 'particles' to allow for a potential P content of these.

- Line 316-319 – I agree that the use of labelled and unlabelled reagents is a way to assess potential error due to hydrolysis of organic P compounds. However, are the authors suggesting that this should be incorporated as standard practice in all _18OP analyses? How feasible is this? If not feasible, then we need a protocol that we can be certain does not risk hydrolysis of organic P compounds. As I comment on above, I'm not convinced that this is the case with the use of MagIC for freshwater samples.

We do it routinely for HCl. The limitation is giving by the size of the sample. Alternatives are not really existing. One possibility could be a physical reduction of the volume, like freeze drying (possible for small volumes) or others. Another possibility would be to apply the DAX resin (it is a resin that adsorb DOP) before the magic step. But this would be costly. Other possibilities are not existing at the moment, at lest for what I know.

- Line 425-428 – Why is washing to remove Cl- so important here, given that Step XII eliminates excess Cl- prior to Ag3PO4 precipitation by adding AgNO3?

It is always important to remove Cl-. The more Cl we have, the more AgNO3 we have to add and this could then entrain the formation of AgO in the final product.

- Line 449-452 – similar to Goldhammer et al. (2011), our observations also suggest AG50WX8 resin may generate a pink colour in solution, despite resin preparation using HNO3 and DD-H2O followed by immediate use. However, in contrast to Goldhammer, we did not observe any adverse effect on Ag3PO4 precipitation, although we follow the

precipitation stages in the McLaughlin et al. (2004) method, rather than the Tamburini et al. (2010) method.

Thank you for this additional information, we will include it in the updated manuscript.

- Line 504-505 – which O-bearing compounds do the authors expect to be removed through this washing step?

The water washing of the Ag3PO4 crystals is important because you eliminate nitrates from the previous steps. If nitrate remains, you have an extra source of oxygen, which is visible then in the Oxygen yield of the samples.

- Line 525 – prior to this step, I assume crystals need to be removed from the filter papers and added to plastic vials? What experience do the authors have with this process, for example are the Ag3PO4 crystals difficult to handle due to static electricity?

Absolutely, it is important here to be very careful. We have experienced an adverse affect from static electricity when transferring Ag3PO4 crystals to silver timbles using plastic spatulas. I would avoid plastic spatulas, use metallic as few problems have been experienced with metallic spatulas. For sure, this is a step where you lose material.

- Line 535 – one of the risks of a multi-stage purification/precipitation process is the loss of phosphate in solution or of P in a solid precipitate during processing, for example if MAP hasn't fully dissolved (lines 431-432). Do the authors have any feeling for how significant these 'losses' of P may be within their protocol? Would this ever lead to an insufficient mass of Ag3PO4 being generated for analysis, despite an apparently sufficient mass of P being present in the initial freshwater sample?

This could happen. Unfortunately, the chemistry of the samples is influencing the success of the purification. This is why it is important to pay always a lot of attention on each step and also to know what the samples are made of.
* * *
469, 2019.

---

## Author Comment (AC3) · 17 Jan 2020

3. Technical corrections Line 245 – depends on, rather than deviate? Line 270 – wording of '. . ..and thy all utilizes a pH. . .' needs to be clarified. Line 277 – Filter, rather than filtrate? Check this throughout the manuscript. Line 339 – present, rather than precent? Line 343 – modified from, rather than modified by? Line 470 – filtered, rather than filtrated? Check throughout the manuscript. Line 490 – important, rather than impotent? Line 494 – not sue what is meant by easterly? Interactive comment on Hydrol. Earth Syst. Sci. Discuss., https://doi.org/10.5194/hess-2019- 469, 2019.

In the revised manuscript we will address all of these technical corrections. Line 245-

depends on Line- 270 this all changes. Line 277- filter, yes we will check it. Line 339- present Line 343- modified from Line 470- filtered Line 490 - important , Line 494 - easily. We will check and double check these typos in the revised document.

———————————————————

---

## Author Comment (AC4) · 17 Jan 2020

Anonymous Referee #3 This MS essentially reads as a long protocol on a method to process freshwater samples to recover sufficient phosphate to allow for isotopic analysis of 18O:16O ratios. While there are a number of published approaches to do so for soils, marine waters, vegetation and other P bearing compounds, the lack of an approach for freshwaters has challenged researchers interested in developing a better understanding of P sources and cycling in

freshwater systems. While I find the MS a bit long and perhaps dry in terms of discussion, I note that many of the observations documented within are similar to experiences that myself and others have observed when trying various approaches. Importantly, commentary on what does not work within a given step is a valuable contribution, particularly when one considers time and financial costs to go from large sample volumes to a few grains of silver phosphate. For those of us working on small budgets, such information is extremely important. In the interests of shortening the MS, perhaps the authors might consider a tabular format which links agreement/disagreement with prior literature support as well as known issues with P recovery? Specific comments

Firstly I would like to thank the reviewer for their helpful comments and their appreciation of our primary motivations in putting together a paper such as this in order to attempt to save researchers considerable time, money and effort in processing and analyzing samples for analysis of oxygen isotopes of inorganic phosphate.

Lines 40-52 – given that in freshwaters, P is most often limiting, some indicate of the range of P found in freshwaters might be useful for readers. Further, some discussion of the relatively likelihood of where 18O-PO4 could be expected to function as a tracer (i.e. low light, eutrophic) vs simply an indication of probably P-recycling pathways (high light, oliogotrophic) might also be useful.

Good point, we will include this in the corrected version of the manuscript.

Line 145-6 – we have used volumes of up to 200 L for low P lakewater samples as typical concentrations are _ 0.05 uM. 0.4 uM is very high for many freshwaters in the northern hemisphere unless significantly affected by human activity or high natural sources of P.

Line 173-75 While in principle this is impractical, we have used sequences of 1 um and 0.2 um polypropylene cartridge filters under low pressure to filter >150 L down to 0.2 um with little problem.

I agree, this is often quite impractical and a filter of nylon polymer 10 micron pore size screen is often useful to act as an initial screen to separate some particles and agglomerated particles. We will include your suggestion as you outlined as an alternative approach (see also R#2 in filtration).

Lines 180-82 – we have found that NaOH loading (as % v:v) has a strong influence on recovery of DOP/DOM. Step V – In our experience, working in high C:P waters, NaOH loadings of _ 1 % v:v are required to completely remove Pi from supernatant in presence of highly colored brucite floc. Furthermore, once we have reduced to a working volume (ca 250 mL) the flocs become more intensely colored rather than less intensely colored as described here. Further, as described in Step VI, after acid dissolution at this stage, the is a darkly colored precipitate that does not dissolve. While we have not determined the nature of this precipitate, loss on ignition when combusted after lyophilisation is near 80% suggesting some acid insoluble organics. We have used columns (similar to Coleman 2002) to "clean" this brownish color from samples with good success. C2

If the final solution from the last dissolution of the brucite is colored, many times I have filtered before proceeding on the next step.

Lines 287-302 – It should be worth adding that significant work on MAGIC in freshwaters has been done with respect to quantitation of Pi (i.e. Anagnostu and Sherrell 2008) and has identified other interfering compounds worth mentioning. Interactive comment on Hydrol. Earth Syst. Sci. Discuss., https://doi.org/10.5194/hess-2019- 469, 2019.

Certainly, we will include this in the corrected manuscript and it will be a very useful resource for researchers.
* * *

---

## Author Comment (AC5) · 17 Jan 2020

Interactive comment on"Analysis of oxygen isotopes of inorganic phosphate ($\delta$18Op) in fresh water: A detailed method description"by Catharina Simone Nisbeth et al.

Anonymous Referee #4

The manuscript by Nisbeth and colleagues describes a protocol to extract phosphate from freshwater samples for the analysis of phosphate oxygen isotope signatures($\delta$18Op). There are several protocols for this extraction to date, all having been developed for different sample matrices
(seawater, freshwater, soil solution, waste water et cetera). One aim of the paper is to critically review these protocols for their adapt-ability to freshwater samples (which would be very welcome), and another is to provide the reader with a detailed method description on how to convert freshwater phosphate to IRMS-grade silver phosphate (which would be even more welcome). However, the authors have decided to interweave both parts (the method review andC1the protocol), and therefore its usability as a concise method description is limited. Firstly I would like to thank the reviewer for the useful comments and suggestions. In regard to the point made that the usability of the technical note as a concise method description , there must be some appreciation that in order to give some level of context and background to the many problems encounter by researchers applying the method a level of description is required to give that detail. As another reviewer outlined, describing some pitfalls or mistakes is very useful as it can save the reader a lot of time , effort and money in the technique application. It is also on the lengthy side for what I would expect from a technical note. Thus, it would strongly benefit from a better organisation; maybe a structure that separates the review part from the description of the (novel) method. Granted this is a valid point and we will work to be more succinct and concise in our descriptions in the corrected document. These formal issues aside, I am missing a robust, data-backed method validation. What have the authors undertaken to verify the integrity of the method? There is unfortunately no data on, for example, phosphate recovery in the individual steps, $Ag_3PO_4$yields,$\delta18Op$ conservation. I consider this absolutely mandatory for a method pa-per (as a HESS technical note or elsewhere), and I strongly suggest that the authors include such evidence in a thorough revision before further consideration.

This is a valid point and we will make reference to method validation through phosphate recovery by using a reference report in the corrected manuscript. Colleagues will also send me the estimate losses with the Magic steps which were estimated to be around 10-15%, but we will be more precise in the corrected manuscript. We will add this info also in the revised version of the paper.

[Figure]

Other suggestions: Title. A little misleading, because the actual measurement of d18OP is not part of the paper (Analysis of oxygen isotopes of inorganic phosphate ($\delta$18Op) in freshwater: A detailed method description) – more apt would be something like "Purification of silver phosphate from freshwater samples for the determination of$\delta$18Op".

We can reformulate the title of the manuscript in the corrected document to something similar to what is suggested here, focusing on the processing procedure. . However, as noted previously, the revised paper with added experimental data might call for yet another title – we will decide later.

Line 199 and elsewhere. Mg-brine is probably not a common term, why not using-MgCl2? That said, I would avoid adding chloride anyway because one will have to get rid of it before Ag3PO4 precipitation.

Mg-Brine is a term sometimes used but we can change this to concentrated MgCl2 as suggested in the corrected manuscript.

Quality of the figures. The photos appear rather blurry and the presentation of lab equipment/vials/glassware would benefit from better organization. Some photos maybe redundant (e.g. Fig 10)

We can endeavour to improve the photos and we will organise them better. I would like to thank the reviewer for the useful comments.